# *Lactobacillus plantarum* KAD protects against high-fat diet-induced hepatic complications in Swiss albino mice: Role of inflammation and gut integrity

**Supriyo Ghosh**[1], **Amlan Jyoti Ghosh**[1], **Rejuan Islam**[1], **Sagar Sarkar**[1,2], **Tilak Saha**[1]*

**1** Immunology and Microbiology Laboratory, Department of Zoology, University of North Bengal, Siliguri, West Bengal, India, **2** Department of Zoology, Siliguri College, Siliguri, West Bengal, India

* tilaksaha@nbu.ac.in

**Data Availability Statement:** All relevant data are within the manuscript and its Supporting Information files.

## Abstract

Hepatic complications are the major health issues associated with dietary intake of calorie saturated food e.g. high-fat diet (HFD). Recent studies have revealed the beneficial effects of probiotics in HFD fed mice with hepatic complications. Some probiotic Lactic acid bacteria (LAB) e.g. *Lactobacillus plantarum* have drawn our attention in managing hepatic complications. Here, we aim to elucidate the protective effects of *L. plantarum* KAD strain, isolated from ethnic fermented food 'Kinema' in HFD-fed mice as, a preventive approach. Eighteen Swiss albino mice were equally divided into 3 groups: Normal Diet (ND), negative control (HFD), and HFD-fed with oral *L. plantarum* KAD supplementation (LP). All the experimental groups were subjected to specific diet according to grouping for eight weeks. After completion of the regime, subjects were anesthetized and sacrificed. Organs, blood, and fecal samples were collected and stored appropriately. Physical indices, including body weight gain, organ co-efficients were calculated along with assessment of glycemic, lipidomic, hepatic, oxidative stress, inflammatory, and histological parameters. Gut microbiota analysis was performed using 16s V3-V4 fecal metagenomic profiling, and sequencing were done using Illumina Miseq system. Oral administration of *L. plantarum* KAD is found to significantly ($p<0.05$) restore metabolic health by normalizing glycemic, lipidomic, hepatic parameters, oxidative stress and inflammatory parameters. Moreover, LP group (7.08±0.52 mg/g) showed significantly ($p<0.001$) decreased hepatic triglyceride level compared to HFD group (20.07±1.32 mg/g). *L. plantarum* KAD improved the adipocytic, and colonic histomorphology with significantly better scoring pattern. LP group (1.83±0.41) showed a significantly ($p<0.001$) reduced hepatic score compared to negative control group (5.00±0.63), showing reduced hepatosteatosis, and immune infiltration. The strain modulated gut health by altering its microbial composition positively towards normalization. *In conclusion, the results of the experiment suggest that prophylactic L. plantarum KAD administration has beneficial* effects on the onset of HFD induced hepatic complications in mice. Further studies are needed, on this strain for its clinical use as dietary supplement.

**Funding:** The author(s) received no specific funding for this work.

**Competing interests:** The authors have declared that no competing interests exist.

## 1. Introduction

The global wave in obesity rates has reached alarming proportions, with profound implications for public health [1]. There is also growing global health concern, on hyperglycemia and associated hepatic complications e.g. non-alcoholic fatty liver disease (NAFLD) [2]. In case of disease progression of any metabolic disorder, inflammation plays a significant role. In fact, inflammation is closely linked to the gut integrity, where disruption in the gut barrier can lead to systemic inflammation and metabolically disturbed state [3]. Moreover, obesity, characterized by excess adipose tissue is a major contributor to the inflammation, further exacerbating insulin resistance (IR) and metabolic dysregulation [4]. This IR results into the development of hyperglycemia and associated issues [5, 6]. Notably, gut inflammation and increased gut permeability are connected to hepatic damage, leading to NAFLD and non-alcoholic steatohepatitis (NASH) like conditions [7]. This complex interplay of thefactors points towards the urgency of innovative therapeutic interventions. The multifaceted nature of these disorders involves intricate pathways that interlace obesity, IR, and hepatic dysfunction [8]. Hyperglycemia further heightens the intricate web of metabolic dysregulation [5, 6]. The liver stands as a crucial point in this pathophysiological cascade. This ultimately results into hepatic injuries ranging from non-alcoholic fatty liver (NAFL) to NASH involving excess fat deposition i.e. steatosis [9]. The convergence of these complications substantially burdens healthcare systems globally, necessitating a comprehensive understanding and effective therapeutic strategies. Obesity-associated metabolic disorders are treated often with limited efficacy and that also with side effects. Lifestyle interventions, pharmaceuticals, and surgical approaches have been implemented, but their success is tempered by factors such as patient adherence, tolerability, and long-term sustainability [10]. As the global burden of obesity-related metabolic disorders continues to escalate, the exploration of alternative therapeutic avenues becomes imperative. Probiotics are beneficial microorganisms that confer health benefits to the host. Preventive approach using probiotics have emerged as a promising frontier in the quest for novel intervention [11]. Among the wide range of probiotic strains, *Lactobacillus plantarum* has shown particular promise in addressing metabolic irregularities [12]. Understanding the elaborate mechanisms by which *L. plantarum* exerts its effects is crucial for exploring its full therapeutic potential [13]. Probiotics have been implicated in modulating the gut microbiota, reducing inflammation, and influencing metabolic pathways, presenting a unique opportunity to combat the intricate network of obesity-related metabolic disorders [14–16]. *Lactobacillus* and *Bifidobacteria* are two major genus which are widely isolated and characterized as probiotics. Scientific study explored that the effectiveness of any strain differs from other which suggests that, the beneficial effects of any bacteria is strain specific [17, 18]. So, the effects of probiotic strain needs to be validated from *in vivo* point of view regardless from which genus it belongs.

Despite the growing evidences supporting potential of probiotics, a knowledge gap persists regarding their specific impact on hepatic health in the context of IR and NAFLD/NASH. The present study aims to evaluate the effectiveness of prophylactic oral supplementation of *L. plantarum* KAD which was isolated from ethnic fermented food 'Kinema' found in Darjeeling, India in high-fat diet (HFD)-fed mice with hepatic complications. This study focuses the strain's potential to manage hepatic, and metabolic complications by assessing glycemic, lipidomic, hepatic, inflammatory, histological, and gut microbiota profiles.

## 2. Materials and methods

### 2.1. Bacterial strains and culture conditions

Previously isolated and characterized *L. plantarum* strain KAD had been used in this study. The 16s rDNA sequences of the strain had been submitted to GenBank (Accession no.

OQ306564) for public availability [19]. The strain was grown in MRS Broth (Hi-Media, India) at 37 ˚C. For oral administration to mice, bacterial cell culture of the strain mentioned above was prepared at the concentration of $2\times10^{10}$ CFU/mL PBS.

## 2.2. Animal grouping and diet

Six-week-old male Swiss albino mice supplied by Chakraborty Enterprise, Kolkata, India, with proper authorization (Regd. No. 1443/PO/BT/s/11/CPCSEA) were maintained in a humidity and temperature-controlled environment (22 ± 1 ˚C and 45 ± 10%) on a 12 h light/dark cycle. After one week of acclimatization with regular pellet feeding, mice were divided into three groups (n = 6 per group): regular pellet diet-fed normal control (ND), HFD-fed negative control (HFD), HFD-fed *L. plantarum* strain KAD treated (LP) groups. The entire experimentation spanned eight weeks according to previous studies. Mice of ND and HFD groups received regular pellets and high-fat diet, respectively. At the same time, mice of LP group are fed the same HFD regime and additional supplementation of 500 µL of the *L. plantarum* strain KAD culture mentioned in Para 2.1. [20, 21].

## 2.3. Sample collection

After the end of the experiment fasted animals were anesthetized using sodium pentobarbital (50 mg/kg; i.p.). Under anesthesia experimental subjects were euthanized by cervical dislocation. Blood was collected by cardiac puncture, from which a batch of samples was kept in EDTA-coated vials, and another set of samples were kept for serum collection. Furthermore, organs like liver, pancreas, perigonadal white adipose tissue (pgWAT), and colon were collected and stored at -20˚C temperature. Fecal samples were collected in vials employing aseptic methods and stored at -20˚C temperature for further use. To alleviate the suffering of experimental mice we have followed some necessary steps following Committee for Control and Supervision of Experiments on Animals (CCSEA) formerly known as, The Committee granted this approval for the Purpose of Control and Supervision of Experiments on Animals (CPCSEA) guidelines: 1. Obtain ethical approval (Reference number IAEC/NBU/2022/28; dated 23.09.2022) from the Institutional Animal Ethics Committee (IAEC) at the University of North Bengal, West Bengal, India and justify animal use, 2. Proper anesthesia and euthanasia methods were followed, 3. Appropriate housing with environmental enrichment were used to reduce stress, 4. Continuous monitoring of animals were done.

## 2.4. Body weight and organ weight gain indices

Percent body weight (BW) gain was calculated according to a standard formula [22].

$$\% \; BW \; Gain = \left(\frac{Final \; body \; weight - Initial \; body \; weight}{Initial \; body \; weight}\right) \times 100$$

Organ co-efficients were calculated to determine the organ weight gain using standard formula [23].

$$Organ \; co-efficient = \frac{Organ \; weight}{Final \; body \; weight} \times 100$$

## 2.5. Glycemic parameters

**2.5.1. AUC (Area Under Curve) $_{Glucose}$ determination.** Blood glucose (BG) levels were measured using a handheld glucometer (Dr. Morepen BG-03, Morpen Laboratories Ltd., India) at 0, 30, 60, and 120 minutes after administering an oral dose of glucose at 2g/kg body

weight. Finally, AUC$_{Glucose}$ was calculated according to standard protocol [24]. AUC$_{Glucose}$ values were derived using the following formula,

$$AUC_{Glucose}(mg\ h/dL) = \frac{BG_{0\ min} + BG_{30\ min} \times 2 + BG_{60\ min} \times 3 + BG_{120\ min} \times 2}{4}$$

**2.5.2. Fasting insulin, glycosylated haemoglobin percentage (GHb%), and homeostatic model assessment for insulin resistance (HOMA-IR) estimation.** Fasting serum insulin and GHb% were measured using standard kits (Mouse Insulin ELISA Kit, MyBioSource, USA; Glycosylated Hemoglobin Kit, Coral Clinical Systems, India). HOMA-IR was calculated according to the standard protocol [25], using the following formula.

$$HOMA - IR = \frac{Fasting\ BG(mmol/L) \times Serum\ insulin\ level(mU/L)}{22.5}$$

## 2.6. Lipidomic parameters

**2.6.1. Biochemical analysis.** Standard kits (Coral Clinical Systems, India) were used to evaluate the lipid profile of the subjects. Biochemical analysis includes triglycerides, cholesterol, and high-density lipoprotein cholesterol (HDL-C). Subsequently, Friedewald's formula was used to determine low-density lipoprotein cholesterol (LDL-C) and very-low-density lipoprotein cholesterol (VLDL-C) levels [26].

**2.6.2. Histological analysis.** After the animal sacrifice, pgWAT was washed in 0.9% normal saline and treated with 4% formalin solution. Then, they are subjected to a gradient of downgrade and upgrade alcoholic series, followed by infiltration and tissue embedding in paraffin. Standard protocol was followed for hematoxylin and eosin (HE) staining [27]. Histological slides were examined using a light microscope (Olympus Magnus, Japan), and the mean adipocyte area was calculated using Magvision Image Analysis Software at 40× objective.

## 2.7. Hepatic functional parameters and oxidative stress

**2.7.1. Biochemical analysis.** Serum gamma-glutamyl transferase (GGT), alanine aminotransferase (ALT), and aspartate aminotransferase (AST) levels were quantified using standard kits (Coral Clinical Systems, India).

**2.7.2. Hepatic triglyceride estimation.** Following a standard protocol, triglyceride was extracted from hepatic tissue [28]. 75 mg of tissue was homogenised in 1.5 mL of chloroform/methanol (SRL, India) (2:1) solution and kept for 2 hours in a shaker incubator at room temperature. Moreover, 150 μL of 1M $H_2SO_4$ (SRL, India) was added to the prepared solution and then centrifuged at 2000 g for 20 mins. The bottom layer was collected and mixed with an equal volume of 1% Triton X-100/chloroform (SRL, India) and kept for a whole night. The dried samples were mixed with distilled water and subjected to measure the triglyceride level with a standard colorimetric kit (Coral Clinical Systems, India).

**2.7.3. Hepatic oxidative stress.** 10% hepatic tissue homogenate was prepared by homogenizing the tissue in phosphate buffer (PB) (50 mM; pH 7.4) (SRL, India). Subsequently, the supernatant was collected after centrifugation of the tissue homogenate at 1000 g for 10 mins at 4°C, and later, it was used for further tests [29]. To carry out the hepatic superoxide dismutase (SOD), relative glutathione (GSH), malondialdehyde (MDA) equivalent and, catalase levels were quantified using standard protocols [30].

**2.7.4. Histological analysis.** HE staining of histological sections of hepatic tissue were prepared and observed using the previously described protocol in para 2.6.2. Hepatic scoring was performed according to standard protocol with slight modifications [31].

## 2.8. Colonic integrity and oxidative stress

**2.8.1. Colonic integrity.** To assess the colonic integrity, serum lipopolysaccharide-binding protein (LBP) level was measured using a standard ELISA kit (Elabscience, USA).

**2.8.2. Colonic oxidative stress.** Previously described protocols in para 2.7.3 were followed to estimate oxidative stress parameters.

**2.8.3. Histological analysis.** Histological staining procedure was followed as mentioned in para 2.6.2. Histological slides were examined using a light microscope (Olympus Magnus, Japan), and the muscle thickness was calculated using Magvision Image Analysis Software at 10× objective. The macroscopic and microscopic assessment of colonic damage was performed with minor modifications [32, 33].

## 2.9. Inflammatory parameters

To evaluate the hepatic inflammatory status, the serum C-reactive protein (CRP) concentration was measured utilizing a standard ELISA kit (MyBioSource, USA) following the manufacturer's instructions. Furthermore, to assess systemic and tissue specific inflammatory state, markers including tumor necrosis factor-alpha (TNF-α) and interleukin-6 (IL-6) were quantified in serum, colon, hepatic, and adipose tissues. These analyses were performed using standard ELISA kits (BT Laboratories, Shanghai, China), with sample processing and analysis conducted according to the manufacturers' protocols.

## 2.10. Gut microbiota analysis

According to the manufacturer's protocol, bacterial DNA was extracted from fecal contents using QIAamp DNA stool mini kit (Dusseldorf, Germany). The V3-V4 region of 16S rRNA gene was subjected to PCR amplification, and finally, Ampure beads were used to purify the amplicons. Sequencing libraries were constructed using eight additional PCR rounds to add Illumina barcoded adaptors. Moreover, libraries were further purified and quantified using Ampure beads and the Qubit dsDNA High Sensitivity assay kit (Thermo Fisher, USA). Illumina Miseq system (Santiago, USA) was used for sequencing. FastQC (Ver. 0.11.9) and MultiQC (Ver. 1.14) were used for quality control of the raw reads; additionally, adaptor trimming and low-quality reads were removed using TrimGalore (Ver 0.6.5). The fecal microbiota analysis was done by Biokart India Pvt. Ltd. in Bengaluru, India. The sequencing data were deposited in the NCBI Sequence Read Archive (SRA) (Accession number PRJNA1075216).

## 2.11. Statistical analysis

The results were represented as mean ± S.D for 6 mice in each group. Statistical comparisons for all the tests except gut microbiota analysis were performed through one-way analysis of variance (ANOVA) using GraphPad Prism 8 software package (Version 8.0.1), as indicated in each separate experiment with Sidak's multiple comparison tests with α = 0.05. *p* values < 0.05 were considered statistically significant.

## 3. Results

### 3.1. *L. plantarum* KAD supplementation protects mice against HFD-induced body weight and organ weight gain indices

*L. plantarum* KAD supplementation significantly ($p<0.001$) normalized the general physical body indices like percent BW increase, liver coefficient, pgWAT coefficient as well as intestine coefficient when compared with HFD group (**Fig 1A–1D**).

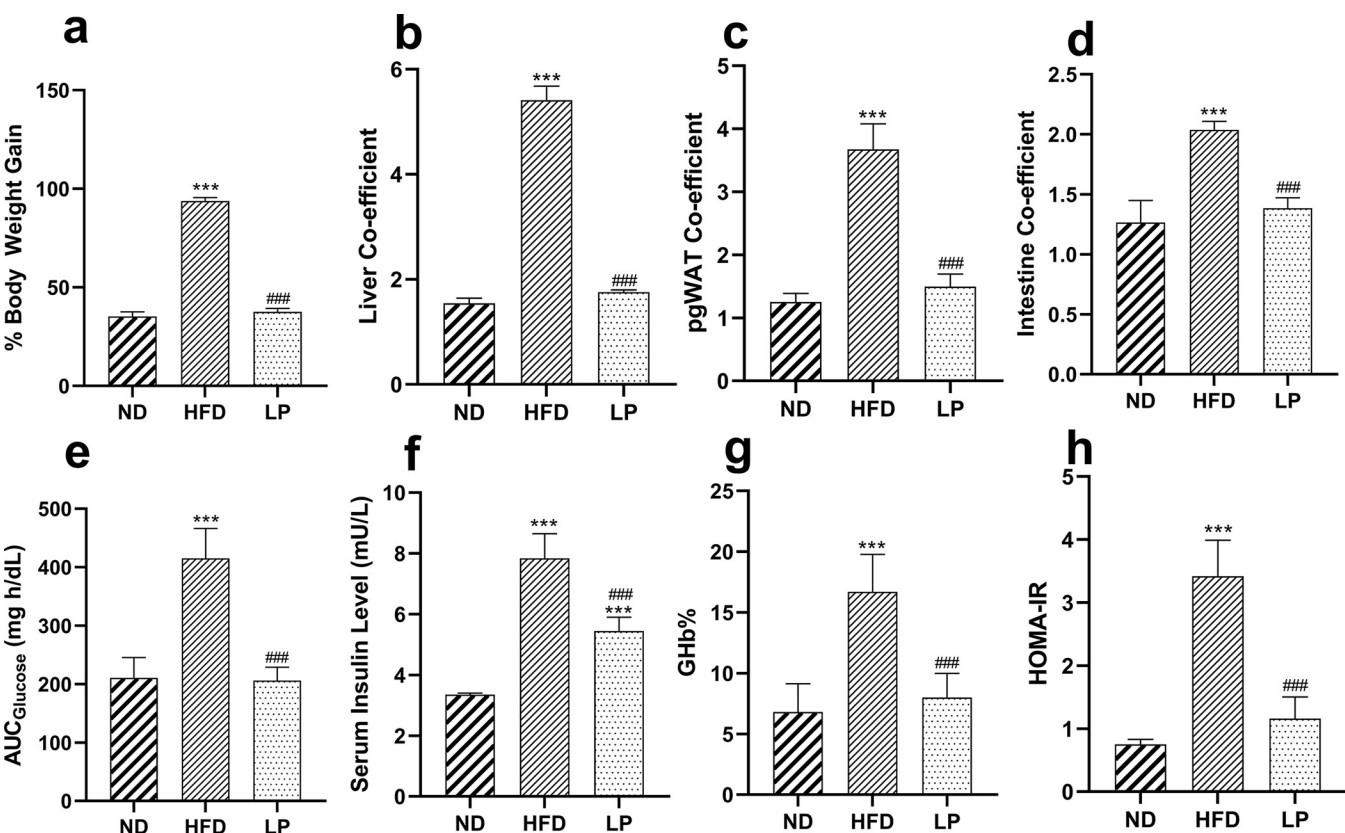

**Fig 1. Eight weeks of prophylactic *Lactobacillus plantarum* KAD supplementation protects mice against HFD-induced body weight, organ weight gain indices, and hyperglycemic state.** Effects of dietary supplementation of *L. plantarum* KAD on **a.** % Body Weight gain, **b.** Liver Co-efficient, **c.** pgWAT Co-efficient, **d.** Intestine Co-efficient, **e.** AUC_{Glucose}, **f.** Serum Insulin Level, **g.** GHb% and, **h.** HOMA-IR among different experimental groups. Differences between experimental groups were analyzed using Sidak's multiple comparison where results are expressed in mean ± SD (n = 6). $^{*}$ $p = 0.033$, $^{**}$ $p = 0.002$, $^{***}p < 0.001$ between ND/HFD and ND/LP, $^{\#}$ $p = 0.033$, $^{\#\#}$ $p = 0.002$, $^{\#\#\#}$ $p < 0.001$ between HFD/LP. ND: Normal pellet diet fed normal control group; HFD: High-fat diet fed negative control group; LP: High-fat diet fed along with prophylactic *L. plantarum* KAD supplemented group; pgWAT: perigonadal white adipose tissue; AUC: area under curve; GHb: glycosylated haemoglobin; HOMA-IR: homeostatic model for insulin resistance.

## 3.2. *L. plantarum* KAD supplementation led to normoglycemic state in HFD-induced mice

LP group showed significantly ($p<0.001$) normalized value of AUC_{Glucose}, GHb%, and HOMA-IR compared with HFD group (**Fig 1E, 1G and 1H**). In terms of the abovementioned parameters, no significant differences were observed when LP group was compared with the ND group. Serum insulin level was significantly reduced in the LP group compared to the HFD group (**Fig 1F**).

## 3.3. *L. plantarum* KAD supplementation suppresses HFD-induced abnormal lipid profile, and pgWAT histology

HFD-treated mice with *L. plantarum* KAD supplementation showed normalized serum triglyceride, cholesterol, HDL-C, LDL-C, VLDL-C level compared to HFD group significantly ($p<0.001$) (**Fig 2A–2E**). The mean adipocyte area was significantly ($p<0.001$) reduced in the LP group in comparison with the HFD group (**Fig 2F and 2G**).

## 3.4. Hepatic functional parameters and oxidative stress

### 3.4.1. *L. plantarum* KAD supplementation lowers hepatic functional parameters in HFD-fed mice. LP group showed significant decrement in serum GGT, AST, hepatic

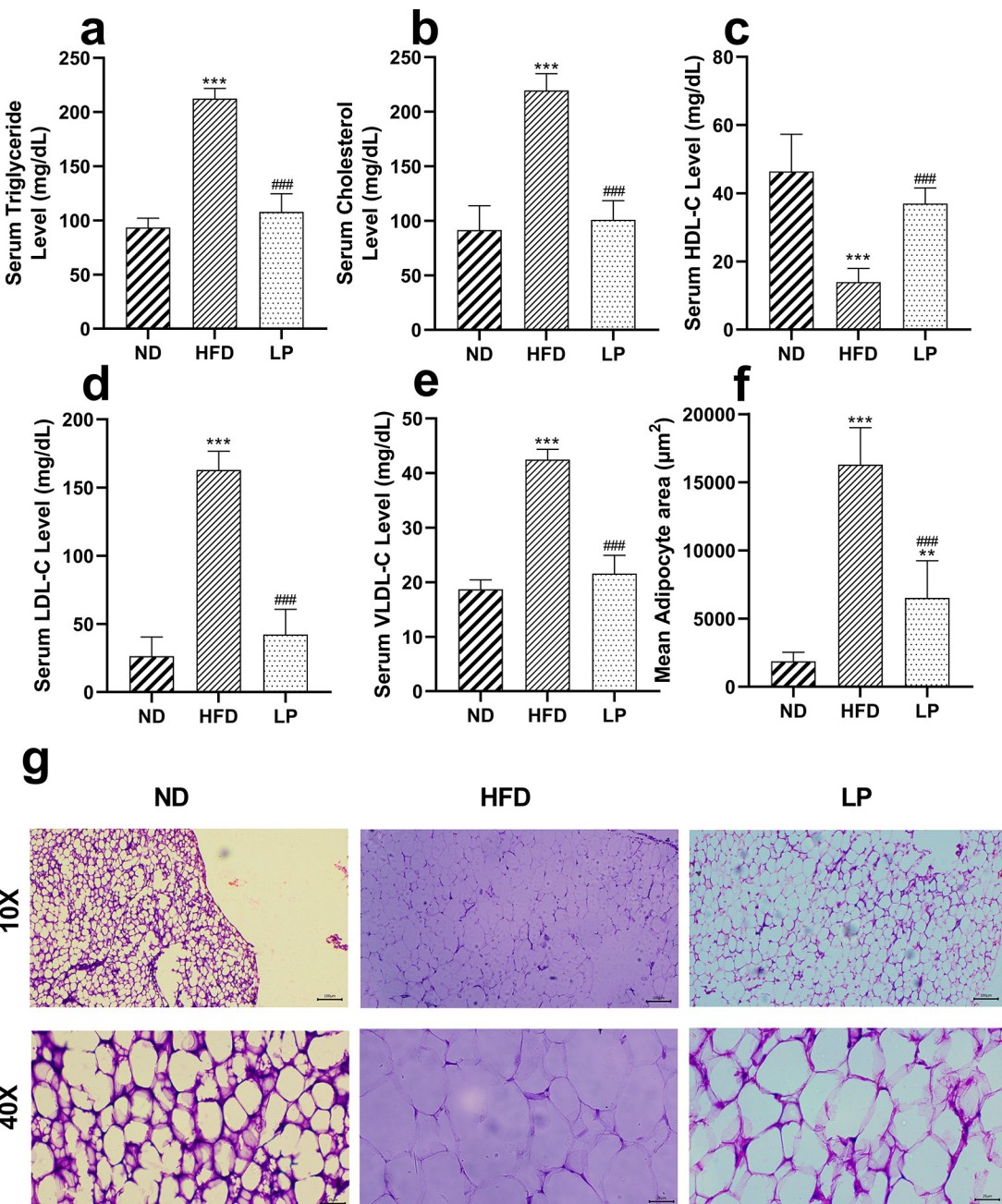

**Fig 2. Eight weeks of prophylactic *Lactobacillus plantarum* KAD supplementation suppresses HFD-induced abnormal lipid profile, and pgWAT histology.** Effects of dietary supplementation of *L. plantarum* KAD on lipidomic parameters, **a.** Serum Triglyceride Level, **b.** Serum Cholesterol Level, **c.** Serum HDL-C Level, **d.** Serum LDL-C Level, **e.** Serum VLDL-C Level, **f.** Mean Adipocyte area and, **g.** pgWAT histology among different experimental groups. Differences between experimental groups were analyzed using Sidak's multiple comparison, where results are expressed in mean ± SD (n = 6). * $p = 0.033$, ** $p = 0.002$, *** $p < 0.001$ between ND/HFD and ND/LP, # $p = 0.033$, ## $p = 0.002$, ### $p < 0.001$ between HFD/LP. Representative photomicrographs were taken from HE-stained histological slides under 10× and 40× objective. ND: Normal pellet diet fed normal control group; HFD: High-fat diet fed negative control group; LP: High-fat diet fed along with prophylactic *L. plantarum* KAD supplemented group; HDL-C: high-density lipoprotein cholesterol; LDL- C: low-density lipoprotein cholesterol; VLDL- C: very low-density lipoprotein cholesterol; pgWAT: perigonadal white adipose tissue.

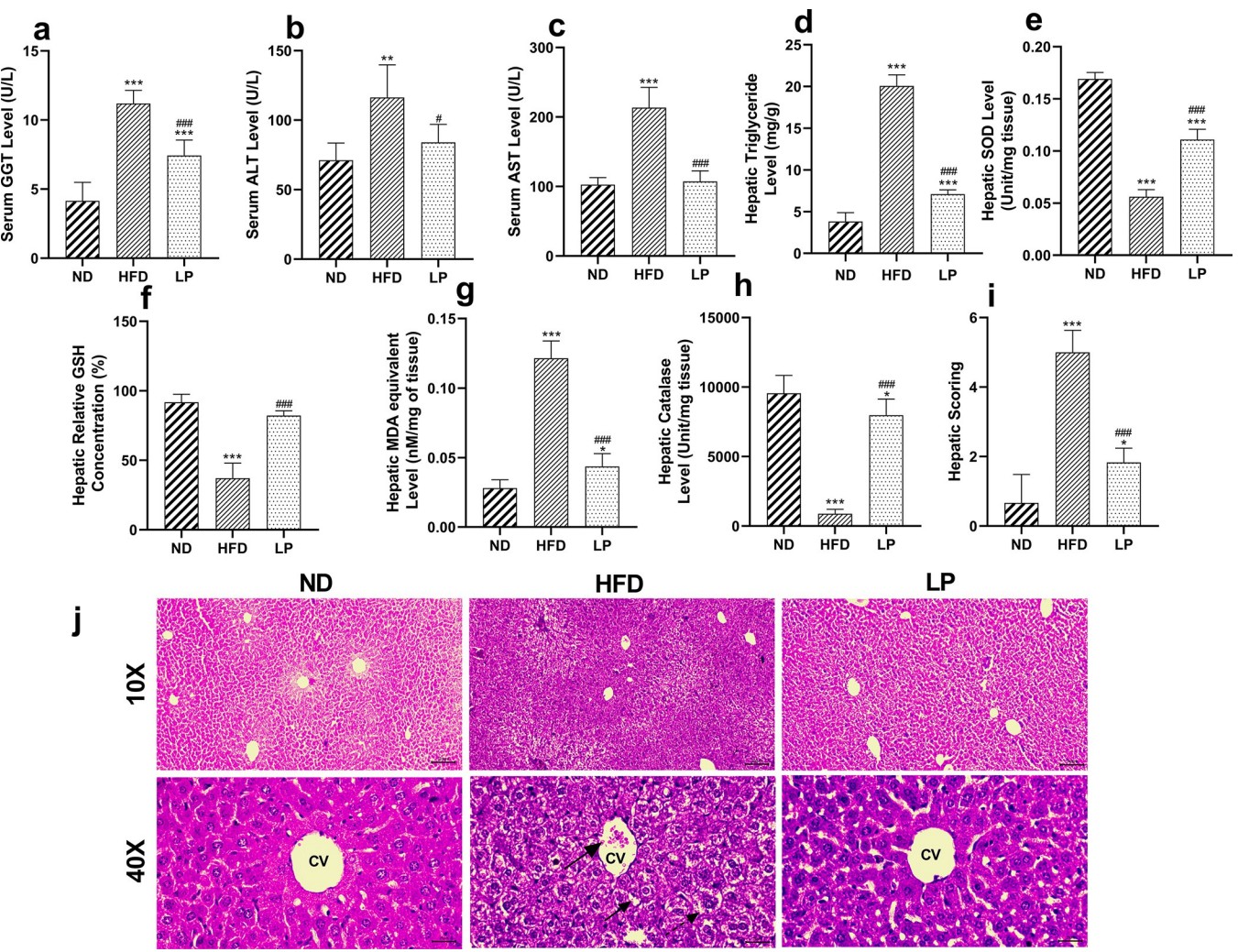

**Fig 3. Eight weeks of prophylactic *Lactobacillus plantarum* KAD supplementation normalizes HFD-induced hepatic functional parameters, oxidative stress, and histology.** Effects of dietary supplementation of *L. plantarum* KAD on **a.** Serum GGT Level, **b.** Serum ALT Level, **c.** Serum AST Level, **d.** Hepatic Triglyceride Level, **e.** Hepatic SOD Level, **f.** Hepatic Relative GSH Concentration, **g.** Hepatic MDA equivalent Level, **h.** Hepatic Catalase Level, **i.** Hepatic Scoring and, **j.** Liver Histology among different experimental groups. Differences between experimental groups were analyzed using Sidak's multiple comparison where results are expressed in mean ± SD (n = 6). * $p = 0.033$, ** $p = 0.002$, ***$p < 0.001$ between ND/HFD and ND/LP, # $p = 0.033$, ## $p = 0.002$, ### $p < 0.001$ between HFD/LP. Representative photomicrographs of were taken from HE stained histological slides under 10× and 40× objective. Arrow with large sized head indicates immune infiltration, arrowhead indicates steatosis, arrow with dashed line indicates increased size of hepatocyte. ND: Normal pellet diet fed normal control group; HFD: High-fat diet fed negative control group; LP: High-fat diet fed along with prophylactic *L. plantarum* KAD supplemented group; GGT: gamma-glutamyl transferase; ALT: alanine aminotransferase, AST: aspartate aminotransferase; SOD: superoxide dismutase; GSH: reduced glutathione; MDA: malondialdehyde; CV: central vein.

triglyceride level compared with HFD group with the *p* value <0.001, = 0.03, <0.001 and <0.001 respectively (**Fig 3A–3D**).

**3.4.2. *L. plantarum* KAD supplementation tends to restore the hepatic anti-oxidant status of HFD-fed mice.** LP group restored the overall antioxidant enzyme profile compared to the HFD group. Hepatic SOD level and catalase level were significantly ($p<0.001$) increased compared to HFD group (**Fig 3E and 3H**) whereas, hepatic relative concentration of GSH was significantly ($p<0.001$) normalized in the LP group compared to HFD group (**Fig 3F**). Hepatic MDA equivalent level was significantly ($p<0.001$) decreased in the LP group than the HFD group (**Fig 3G**).

**3.4.3. *L. plantarum* KAD supplementation reduces steatosis, and improves hepatic histomorphology.** In the HFD group steatosis, immune infiltration, swollen hepatocytes with distorted histology was observed compared to the ND group. In the LP group histoarchitecture showed restoration of the histomorphology (**Fig 3J**). In the LP group hepatic scoring value was significantly ($p<0.001$) reduced compared to the HFD group (**Fig 3I**).

## 3.5. Colonic integrity and oxidative stress

**3.5.1. *L. plantarum* KAD supplementation improves colonic permeability, oxidative stress, and histomorphology in HFD-fed mice.** The level of serum LBP was found to be significantly ($p<0.001$) higher in the HFD group than in the LP group, suggesting the beneficial role of *L. plantarum* KAD supplementation (**Fig 4A**).

**3.5.2. *L. plantarum* KAD supplementation tends to restore the colonic anti-oxidant status of HFD-fed mice.** Overall, colonic antioxidative enzymes were found to be restored significantly in the LP group compared to the HFD group. The colonic SOD, relative concentration of GSH, and catalase levels were found to be significantly ($p<0.001$) higher in the LP group than the HFD group (**Fig 4B, 4C and 4E**). Whereas the LP group showed significantly ($p<0.001$) lowered colonic MDA equivalent level compared to the HFD group (**Fig 4D**).

**3.5.3. *L. plantarum* KAD supplementation improves colonic histomorphology.** Macroscopic evaluation for colonic damage showed that there was no significant difference between the ND group, and LP group, whereas the HFD group showed marked ($p = 0.002$) increase in the scoring pattern compared to the ND group (**Fig 4F**). Similar pattern was also followed in case of microscopic evaluation of colonic damage. The LP group showed significantly ($p<0.001$) reduced microscopic score compared to HFD group (**Fig 4G**). Histological analysis showed elevated level of histological damages including goblet cell depletion, reduced muscle thickness, immune infiltration in the HFD group, whereas the LP group showed normalized histoarchitecture like the ND group (**Fig 4H**).

## 3.6. *L. plantarum* KAD supplementation suppresses HFD-induced inflammation in serum, liver, colon, and adipose tissue

The LP group showed significantly ($p<0.001$) restored serum CRP level in comparison with the HFD group (**Fig 5A**). In the serum, TNF-α level of the HFD group showed a significant ($p<0.001$) increase compared to the ND group. In the LP group serum TNF-α level was although higher than the ND group but, remained significantly ($p<0.001$) reduced compared to the HFD group (**Fig 5B**). The mean colonic TNF-α level was significantly ($p<0.001$) increased in the HFD group compared to the ND group. In the LP group, the colonic TNF-α level was significantly ($p<0.001$) reduced compared to the HFD group (**Fig 5F**). The HFD group showed significant ($p<0.001$) increase in the hepatic TNF-α level compared to the ND group. Again, in the LP group TNF-α level of hepatic tissue was significantly ($p<0.001$) reduced compared to the HFD group (**Fig 5D**). In case of serum, liver, and colon the IL-6 levels were significantly ($p<0.001$) higher in the HFD group compared to the ND group. The LP group showed significantly ($p<0.001$) decreased levels of IL-6 in the serum, liver, and colon compared to the HFD group (**Fig 5C, 5E and 5G**). The mean TNF-α levels in the adipose tissue of the HFD group was although increased but, not significant when compared with that of the ND group. Non-significant change in the TNF-α level of adipocyte tissue in the LP group compared to the HFD group was also observed. Similarly, IL-6 level in the ND group was significantly ($p<0.001$) elevated in the HFD group compared to the ND group. The LP group showed a significant decrease ($p<0.001$) in the IL-6 level of adipocyte tissue compared with that of the HFD group (**Fig 5H and 5I**).

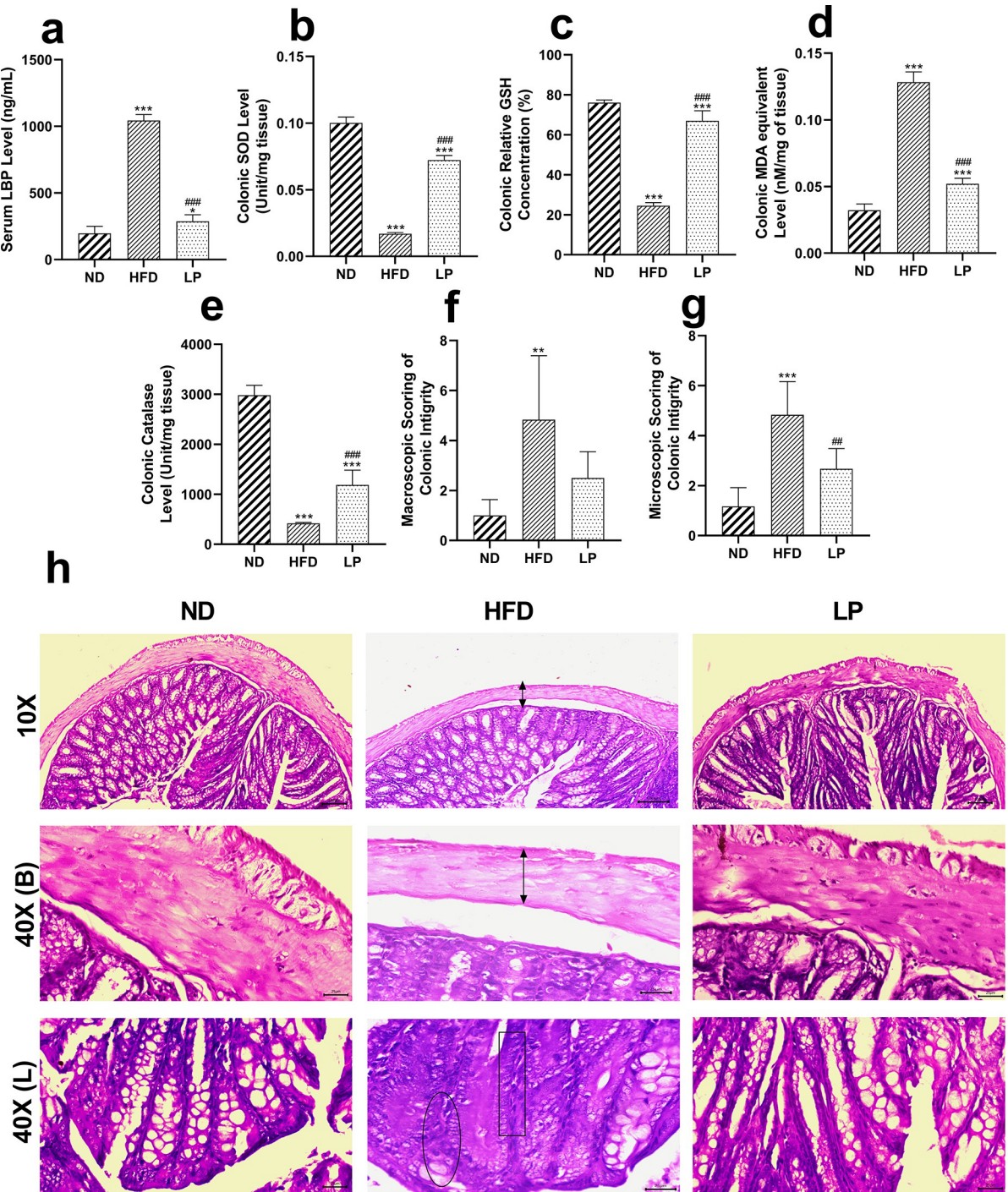

**Fig 4. Eight weeks of prophylactic *Lactobacillus plantarum* KAD supplementation protects mice against colonic permeability, oxidative stress, histological abnormalities.** Effects of dietary supplementation of *L. plantarum* KAD on **a.** Serum LBP Level, **b.** Colonic SOD Level, **c.** Colonic Relative GSH Concentration, **d.** Colonic MDA equivalent Level, **e.** Colonic Catalase Level, **f.** Macroscopic Scoring of Colonic Integrity, **g.** Microscopic Scoring of Colonic Integrity and, **h.** Colon Histology among different experimental groups. Differences between experimental groups were analyzed using Sidak's multiple comparison, where results are expressed in mean ± SD (n = 6). * $p = 0.033$, ** $p = 0.002$, *** $p < 0.001$ between ND/HFD and ND/LP, # $p = 0.033$, ## $p = 0.002$, ### $p < 0.001$ between HFD/LP. Representative photomicrographs were taken from HE-stained histological slides under 10× and 40× objective. Arrow with two heads indicates decreased muscle thickness, rectangular box indicates immune infiltration, circle indicates goblet cell depletion. ND: Normal pellet diet fed normal control group; HFD: High-fat diet fed negative control group; LP: High-fat diet fed along with prophylactic *L. plantarum* KAD supplemented group; LBP: lipopolysaccharide-binding protein; SOD: superoxide dismutase; GSH: reduced glutathione; MDA: malondialdehyde; B: basal side of the colon; L: luminal side of the colon.

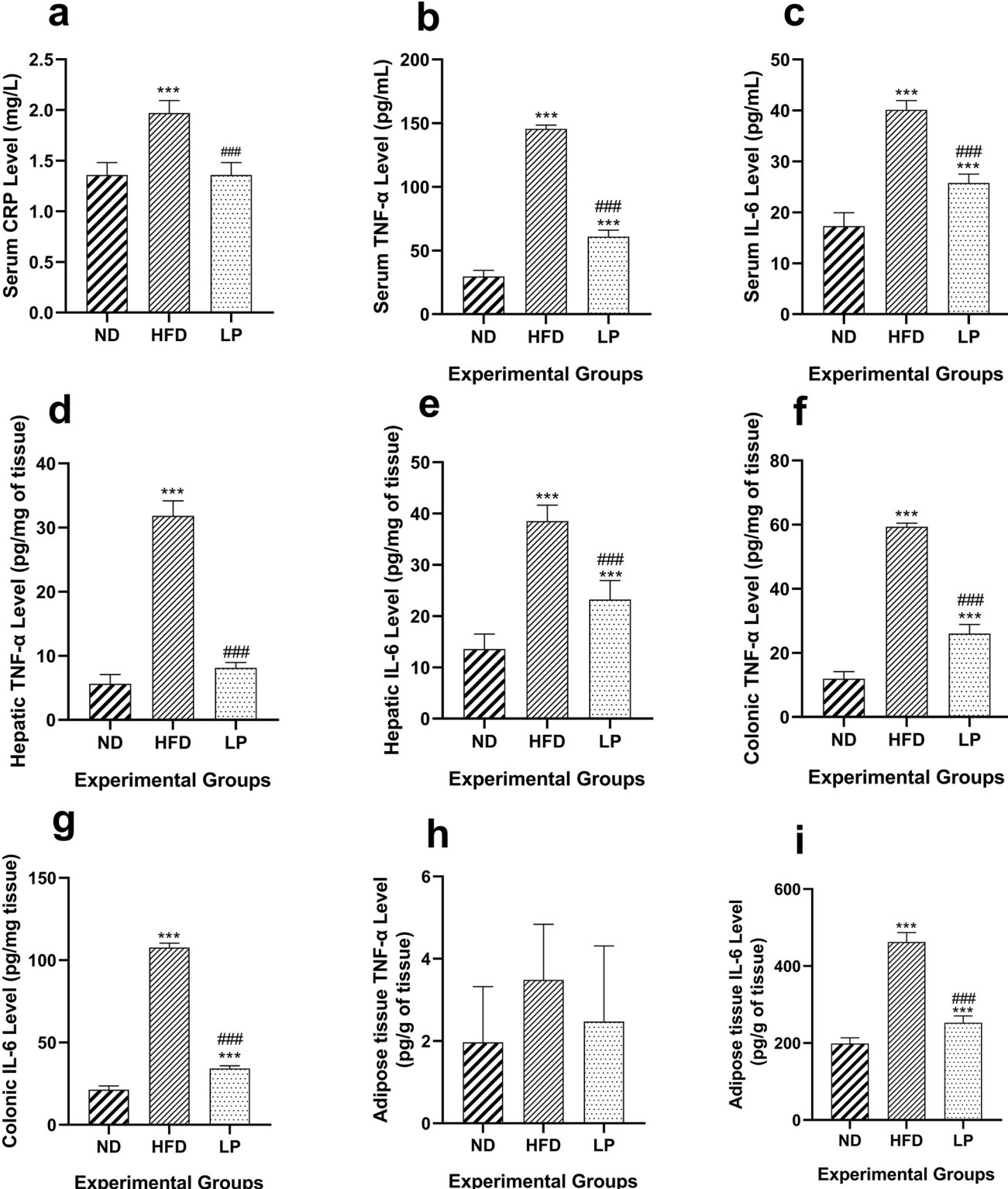

**Fig 5. Eight weeks of prophylactic *Lactobacillus plantarum* KAD supplementation suppresses inflammation in mice.** Effects of dietary supplementation of *L. plantarum* KAD on **a.** Serum CRP Level, **b.** Serum TNF-α Level, **c.** Serum IL-6 Level, **d.** Hepatic TNF-α Level, **e.** Hepatic IL-6 Level, **f.** Colonic TNF-α Level, **g.** Colonic IL-6 Level, **h.** Adipose tissue TNF-α Level and, **i.** Adipose tissue IL-6 Level among different experimental groups. Differences between experimental groups were analyzed using Sidak's multiple comparison, where results are expressed in mean ± SD (n = 6). * $p = 0.033$, ** $p = 0.002$, *** $p < 0.001$ between ND/HFD and ND/LP, # $p = 0.033$, ## $p = 0.002$, ### $p < 0.001$ between HFD/LP. ND: Normal pellet diet fed normal

control group; HFD: High-fat diet fed negative control group; LP: High-fat diet fed along with prophylactic *L. plantarum* KAD supplemented group; TNF-α: tumor necrosis factor α; IL-6: interleukin 6.

### 3.7. *L. plantarum* KAD supplementation exerts a positive influence in the gut microbiota composition of HFD-fed mice

The metagenomic analysis of fecal samples from three experimental groups revealed distinct alterations in the taxonomic composition of the fecal microbiota. At the Phylum level relative abundance of *Bacteroidetes* was increased in the HFD group on the other hand *Firmicutes* were decreased. *Firmicutes/Bacteroidetes* ratio was decreased in the HFD group and, the LP group showed increment of the ratio towards normalization (**Fig 6A**). In case of percent relative abundance among families, *Bacteroidaceae* exhibited a reduction in both HFD and LP group compared to ND. *Lachnospiraceae* demonstrated a marked decline in the HFD group compared to ND, with partial restoration observed in the LP group. *Prevotellaceae* exhibited a notable increase in the LP group compared to ND and HFD groups. *Unclassified Clostridiales* displayed consistent abundance across all groups. *Lactobacillaceae* showed a substantial increase in the LP group compared to HFD group whereas, ND group showed the value of relative abundance. *Unclassified Bacteroidales* exhibited a significant reduction in the LP group compared to the HFD group (**Fig 6B**). Relative abundance of *Clostridium*, *unclassified Desulfovibrionaceae*, *unclassified Enterobacteriaceae*, and *unclassified Muribaculaceae* were increased in the HFD group compared to the ND group whereas, LP supplementation showed restoration of above-mentioned taxa towards normalization. The levels of *Prevotella* and *Lactobacillus* and, *unclassified Lachnospiraceae* were increased in the LP group compared to the HFD group (**Fig 6C**). *Bacteroides sp*. level was increased in HFD, in contrast with it *Bacteroides vulgatus* level was decreased in HFD group compared to ND group. Elevated levels of *Clostridioides difficile*, *Clostridium tyrobutyricum* and *Clostridium sp*. were found in HFD compared to ND group whereas, LP showed more or less similar pattern like ND group. The relative abundance of *Lactobacillus brevis*, *Lactobacillus fermentum*, *Lactobacillus murinus*, *Lactobacillus reuteri*, *Lactococcus lactis*, and *Lactobacillus sp*. showed normalized level in LP compared to ND group (**Fig 6D**).

## 4. Discussion

In this present experiment it was observed that, prophylactic *L. plantarum* strain KAD oral administration showed protective effects in HFD-fed Swiss albino mice with hepatic complications. The strain effectively decreased the hepatic triglyceride accumulation compared to the negative control HFD group. Additionally, hepatic histology was restored in the LP group suggesting the beneficial role of *L. plantarum* KAD. The LP group also showed better physical indices, lipid profile, anti-oxidant status, and inflammatory profile. Furthermore, *L. plantarum* KAD improved the intestinal permeability, and gut microbiota profile towards normalization.

Previous studies have supported that probiotics have immense effects on regulating metabolic disorders [23, 34]. Stabilizing weight gain, body indices, or organ-specific co-efficient is a parameter used to treat metabolic disorders. This study showed that *L. plantarum* strain KAD has the potential in managing such parameters. The LP group showed improved glycemic parameters, including AUC$_{Glucose}$, fasting serum insulin level, GHb%, and HOMA-IR compared to the HFD group, suggesting *L. plantarum* KAD's probable role as a normoglycemic agent. Moreover, the LP group showed an improved lipid profile regarding serum triglyceride, cholesterol, HDL-C, and LDL-C levels compared with the HFD group. Significantly reduced mean adipocyte area from histological evaluation can also be correlated with serum lipid

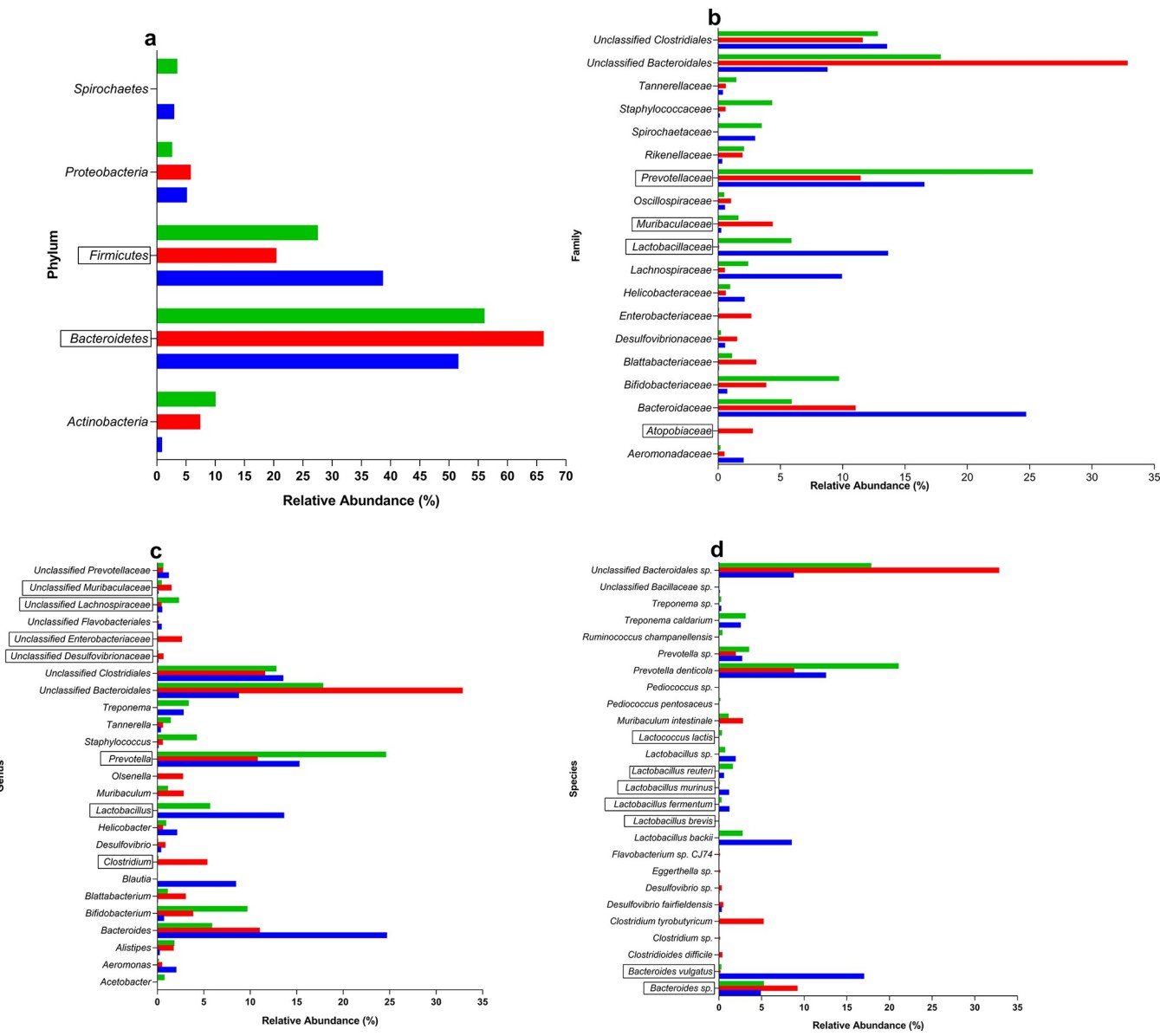

**Fig 6.** Effects of eight weeks of prophylactic *Lactobacillus plantarum* KAD supplementation on fecal metagenomic profile of high-fat diet fed swiss albino mice, at **a.** Phylum, **b.** Family, **c.** Genus and, **d.** Species level among different experimental groups. Blue coloured column indicates ND group, orange coloured column indicates HFD group and, green coloured column indicates LP group. Important taxa co-related with disease were marked within box. ND: Normal pellet diet fed normal control group; HFD: High-fat diet fed negative control group; LP: High-fat diet fed along with prophylactic *L. plantarum* KAD supplemented group.

profile. The mean adipocyte area is closely linked to metabolic disorders. Studies show a positive correlation between increased mean adipocyte area and adverse lipid profiles, contributing to dyslipidemia [20, 28]. According scientific study, in rodent models having hepatic complications with abnormal lipid profiles showed normalizing effects when they were fed with probiotics. This finding also suggests that probiotic supplementation improves lipid profile [35]. Although, the exact molecular mechanism by which *L. plantarum* KAD improves glycemic, and lipidomic parameters need to be explored.

Assessment of GGT, ALT, AST along with lipid profile is important criteria to assess disease progression in an individual with hepatic complications [36]. Increased GGT, ALT, and AST levels are crucial indicators of hepatic damage. The significant reduction in these parameters in the LP group compared to the HFD group suggests a preventive approach of using probiotic supplementation against HFD-induced hepatic damage. *L. plantarum* KAD supplementation improved hepatic histology with significantly decreased hepatic scoring which includes steatosis, immune infiltration and, swollen hepatocyte. These findings align with previous studies indicating the hepatoprotective potential of probiotics [37–39]. The observed decrease in hepatic triglyceride accumulation in the LP group aligned with the hepatic histological findings especially the reduced steatosis. These findings further support the protective role of *L. plantarum* KAD against HFD-induced hepatic triglyceride accumulation. Earlier study demonstrating the ability of *L. plantarum* to modulate lipid metabolism, and reduce hepatic fat accumulation corroborate these results, suggesting the strains's probable preventive role in managing hepatic complication [40]. Modulation of the fatty acid metabolism may be the probable reason behind the decreased hepatic triglyceride accumulation which needs to be validated by further molecular studies.

Antioxidants are unique defense mechanisms against free radicals, including SOD, GSH, and catalase. MDA is frequently employed as a measure of lipid peroxidation and is having a negative impact on both DNA and proteins [40–42]. In this investigation, *L. plantarum* KAD treatment resulted in a considerable decrease in MDA content, and increases in SOD, GSH, and catalase activity. Consequently, *L. plantarum* KAD may function as a possible agent for managing a proper balance of anti-oxidant, and oxidant in cells maintaining a stable state.

Another significant element affecting chronic inflammation under HFD-fed situation is an increase in circulating lipopolysaccharide (LPS) levels due to increased level of intestinal permeability. LBP, the main binding protein for detecting LPS found in circulation. So, increased level of serum LBP is associated with elevated level of LPS under HFD-fed condition denoting leaky gut. Due to leaky gut LPS passes through intestine and reaches liver [43, 44]. LPS which is an endotoxin produced by gram negative bacteria is responsible for hepatic toll-like receptor 4 (TLR-4) activation. This activation initiates several inflammatory cascades, which results into liver damage [45]. Based on the results of this investigation, strain KAD may be responsible for maintaining gut integrity, and mitigating gut-derived endotoxemia mediated the liver damage. This finding aligns with the previous studies but, still needs to be further evaluated.

Elevated gut permeability is one of the results of oxidative stress caused by the intestinal cytochrome P450 2E1 (CYP2E1). According to earlier research, CYP2E1 plays a crucial role in the development of non-alcoholic steatosis carried on by HFD by inducing inflammation, and oxidative stress in the gut. Reactive oxygen species (ROS), oxidative stress, and activation of CYP2E1 metabolism can all have a role in the alteration of intestinal permeability [46–48]. Elevated levels of colonic antioxidant enzymes, and reduced MDA levels suggest that, *L. plantarum* KAD supplementation may suppress the gut permeability by, modulating colonic oxidative stress. Macroscopic and microscopic evaluation for colonic damage shown in previous research also supported these colonic histological findings, suggesting *L. plantarum* KAD's probable protective role to restore colonic health [32, 33].

The inflammatory parameters of the experiment revealed that *L. plantarum* KAD supplementation reduced inflammation not only systemically but, also in metabolically active tissues. The initial inflammatory marker assessed was CRP in the serum, an acute-phase protein produced by hepatic tissues in response to inflammatory stimuli such as metabolic endotoxemia, characterized by compromised gut integrity. Elevated CRP levels trigger a cascade of inflammatory processes throughout the body [49]. This study showed significantly elevated levels of serum inflammatory markers such as TNF-α, and IL-6 in the HFD group, whereas these

cytokines were restored to near-normal levels in the *L. plantarum* supplemented group. These suggest probiotic's anti-inflammatory attributes aligning with the previous scientific study [50]. At the tissue level, colonic inflammatory markers TNF-α, and IL-6 were also significantly elevated in the HFD group but were markedly reduced in the LP group, suggesting a probable anti-inflammatory effect of the probiotic on colonic tissues. This observation correlates with previous research indicating that colonic damage is associated with elevated levels of LBP, and oxidative stress, contributing to a vicious cycle of inflammation, and oxidative damage [20]. Furthermore, TNF-α, and IL-6 are barrier-disrupting cytokines result into compromised gut barrier integrity [51]. In the present study, *L. plantarum* KAD supplementation modulated the colonic inflammatory state, and probably protects the gut integrity by, suppressing such barrier-disrupting cytokines in the gut. Elevated levels of hepatic TNF-α, and IL-6 in the HFD group were observed which, are consistent with previous findings that a HFD induces significant hepatic inflammation. *L. plantarum* supplementation significantly reduced hepatic inflammation, and improved serological hepatic parameters such as- GGT, ALT, AST, and lipid profiles. These indicate *L. plantarum* may have potential role in managing metabolic, and associated inflammatory states. Similarly, in adipose tissues, we found that HFD induced significant inflammation, reflected by IL-6 level, which was reduced in the LP group. This reduction in adipose tissue inflammation correlated with histological findings of smaller adipocyte size and reduced overall body weight gain in the LP group, supporting the notion that obesity-induced inflammation in adipocytes contributes to systemic inflammation and IR [28, 50].

The metagenomic analysis of fecal samples in this animal model experiment elucidated significant shifts in the taxonomic composition of the fecal microbiota among the experimental groups. Metagenomic analysis of gut microbiome from the previous studies corroborates the results of this experiments, suggesting that in the case of hepatic disease, the *Firmicutes* level decreased, and intestinal inflammation led to increased *Bacteroidetes* in the gut [52, 53]. Moreover, decreased *Firmicutes/Bacteroidetes* ratio is associated with metabolic disease which supports the present experimental findings [54]. Notably, the family *Atopobiaceae* and *Muribaculaceae*, associated with ulcerative colitis-like conditions and increased ALT as well as AST level. These families exhibited a substantial increase in relative abundance in the HFD group compared to ND. These suggest a potential link between calorie saturated-diet consumption and the promotion of colitis-associated microbial profiles and hepatic complication respectively [55, 56]. In contrast, the family *Lactobacillaceae* and *Prevotellaceae*, which are related to a healthy hepatic condition, are significantly decreased in the HFD group. However, the LP group led to a considerable increase in these taxa, indicating a potentially beneficial effect on hepatic health [57]. The notable increase in *Lactobacillus*, *Prevotella*, and *unclassified Lachnospiraceae* abundance in the LP group suggests a positive influence of *L. plantarum* KAD, aligning with previous studies highlighting its role in attenuating the progression of hepatic complications [58–60]. The observed increase in the detrimental taxa like- *Clostridium*, *unclassified Desulfovibrionaceae*, *unclassified Enterobacteriaceae*, and *unclassified Muribaculaceae* in the HFD group negatively impact the mice hepatic health, as it is associated with the occurrence and development of hepatic damage [60–62].

The increased abundance of *Bacteroides sp.* in the HFD group, known to induce systemic inflammation [63], underscores the role of dysbiosis in promoting metabolic disturbances. Conversely, the significant reduction in *Bacteroides vulgatus*, associated with lipid metabolism [64], highlights a potential link between dysbiosis and altered lipid homeostasis in HFD-fed mice. Notably, *Lactobacillus brevis*, although present in low abundance, has been reported to ameliorate hepatic disease [53]. Whereas, *Lactobacillus fermentum*, detected at low levels in the ND and, LP group has known to have hepatoprotective properties, and may contribute to mitigating hepatic damage [65]. The observed increase in *Lactobacillus murinus* in the LP group

aligns with its reported capacity to reduce intestinal permeability, thereby potentially attenuating HFD-induced gut barrier dysfunction [66]. Furthermore, the elevated abundance of *Lactobacillus reuteri* and *Lactococcus lactis* in the LP group compared to ND group may confer hepatic protection [67, 68], further emphasizing the potential benefits of *L. plantarum* KAD in the context of liver health. HFD feeding induces gut dysbiosis which was reflected by the metagenomic analysis. Due to the dysbiosis serum endotoxemia had occurred [69]. It may be possible that, along with the leaky gut these endotoxins finally damaged the liver as mentioned earlier. *L. plantarum* KAD may be integrated into therapeutic strategy in case of treating hepatic complications but, detailed molecular studies are required to validate its potential. Overall, strain KAD may be undergone for therapeutic trial with appropriate molecular validation in the context of management of hepatic complications via improving inflammation, gut integrity, and gut microbiota composition.

## 5. Conclusion

This study demonstrates that *L. plantarum* KAD supplementation effectively reduces inflammation, oxidative stress, and metabolic dysfunction in HFD-fed mice with hepatic complications. Histological as well as, biochemical findings support that the probiotic can reduce hepatic fat accumulation in the HFD-fed mice. Prophylactic *L. plantarum* KAD supplementation can also normalize pgWAT, and colonic histoarchitecture with improved scoring pattern. Furthermore, this strain prevents colonic permeability, and modulate the gut microbial composition towards normalization. It may be possible that, *L. plantarum* KAD improves hepatic complications indirectly by improving intestinal permeability, and thereby oxidative stress regulation. These findings support the probable potential role of *L. plantarum* in managing hepatic complications, and HFD-associated metabolic issues. Future research should focus on elucidating the precise mechanisms through which *L. plantarum* exerts its beneficial effects, and explore its applicability in clinical trials for human health improvement.

## Supporting information

**S1 File.**
(DOCX)

## Acknowledgments

The authors express gratitude to the IAEC, Department of Zoology, University of North Bengal for providing necessary ethical permission and extending the infrastructural facility for conducting the research work.

## Author Contributions

**Conceptualization:** Tilak Saha.

**Data curation:** Supriyo Ghosh, Amlan Jyoti Ghosh, Rejuan Islam, Sagar Sarkar, Tilak Saha.

**Formal analysis:** Supriyo Ghosh, Amlan Jyoti Ghosh, Rejuan Islam, Sagar Sarkar, Tilak Saha.

**Investigation:** Supriyo Ghosh, Amlan Jyoti Ghosh, Rejuan Islam, Sagar Sarkar.

**Methodology:** Supriyo Ghosh, Amlan Jyoti Ghosh, Rejuan Islam, Sagar Sarkar.

**Supervision:** Tilak Saha.

**Visualization:** Supriyo Ghosh.

Writing – **original draft:** Supriyo Ghosh.

Writing – **review & editing:** Tilak Saha.

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
