## [Decision Letter · Decision Letter 0]

22 Aug 2024

PONE-D-24-27060Oral supplementation of Lactobacillus plantarum KAD Prevents High-Fat Diet-Induced Hepatic Complications in Swiss Albino Mice: Role of Inflammation and Gut IntegrityPLOS ONE

Dear Dr. Saha,

Thank you for submitting your manuscript to PLOS ONE. After careful consideration, we feel that it has merit but does not fully meet PLOS ONE’s publication criteria as it currently stands. Therefore, we invite you to submit a revised version of the manuscript that addresses the points raised during the review process.

We look forward to receiving your revised manuscript.

Kind regards,

Ahmed Khalafallah Sadeq Rashwan, PhD

Academic Editor

PLOS ONE

Journal Requirements:

3. Thank you for stating the following in your Competing Interests section: On behalf of all authors I disclose that there is no competing interests that could be perceived to bias this work. 

Additional Editor Comments (if provided):

This manuscript number (PONE-D-24-27060) is entitled "Oral supplementation of Lactobacillus plantarum KAD Prevents High-Fat Diet-Induced Hepatic Complications in Swiss Albino Mice: Role of Inflammation and Gut Integrity". The novelty of the manuscript is good although the analyses are not sufficient in this work. This article can be considered for publication after doing the following major revisions:

1. This paper is worthy of publication although the novelty should be addressed clearly in the abstract and introduction

2. The abstract is concise but could benefit from a clearer statement of the study's objectives, methods, main findings, and significance. Start by clearly stating the objective of the study. Briefly describe the methodology, emphasizing the study design and key experimental procedures. Highlight the most significant results, particularly those that demonstrate the impact of Lactobacillus plantarum KAD on hepatic complications. Conclude with a sentence on the study’s implications for future research or clinical practice.

3. The introduction outlines the background and the need for the study but could be more focused on the research gap. Streamline the introduction by focusing on the specific gaps in current research that your study addresses. For instance, emphasize the novelty of exploring Lactobacillus plantarum KAD’s role in hepatic complications within the context of high-fat diet-induced obesity. You might also want to more explicitly state the hypothesis or primary research question.

4. The methods are detailed, but some sections might be too dense. Consider breaking down the methods into more specific subheadings for clarity, such as "Animal Grouping and Diet," "Biochemical Analysis," "Histological Examination," etc. This will make it easier for readers to follow your procedures. Additionally, ensure that all procedures, especially those involving statistical analyses, are described with enough detail to allow replication.

5. The results section is comprehensive but could benefit from more emphasis on the significance of the findings. Instead of just presenting the data, you can enhance the narrative by linking the findings back to the research questions. For instance, when discussing the lipid profile or histological findings, briefly state what these results imply about the effectiveness of Lactobacillus plantarum KAD. Use subheadings to organize the results logically, perhaps under themes like "Glycemic Control," "Lipid Metabolism," "Hepatic Function," etc.

6. The discussion provides a good analysis but can be more tightly linked to the results. Start by summarizing the main findings in a few sentences. Then, critically evaluate these findings in the context of existing literature, discussing how your study confirms, contradicts, or extends previous research. Ensure that the discussion logically follows from the results presented, addressing potential limitations of the study and suggesting areas for future research. Consider framing the discussion around how Lactobacillus plantarum KAD could be integrated into therapeutic strategies for metabolic disorders.

7. The conclusion is currently embedded within the discussion. Create a distinct conclusion section that succinctly restates the main findings, the potential clinical implications, and the next steps for research. This will help to underscore the importance of your work.

8. The choice of inflammatory markers (CRP, TNF-α, IL-6) is appropriate but limited. Including additional markers, such as IL-10 or IL-1β, could provide a more comprehensive picture of the inflammatory response and further substantiate the claims made regarding anti-inflammatory effects.

9. While the study shows that Lactobacillus plantarum KAD has beneficial effects, the mechanistic pathways are not thoroughly explored. The discussion would benefit from more detailed hypotheses or evidence about how the probiotic strain exerts its effects, particularly in relation to gut microbiota modulation and hepatic function.

10. The discussion sometimes overstates the significance of the findings without sufficient qualification. For example, the claims about the potential therapeutic benefits of Lactobacillus plantarum KAD should be tempered by acknowledging the limitations of extrapolating animal model results to human health outcomes.

11. The figures and tables are informative but may need more detailed legends. Ensure that each figure and table have a comprehensive legend that explains all abbreviations and provides enough context for understanding without referring to the text. Consider whether the figures could be reordered or combined to tell a clearer story.

12. The section on gut microbiota analysis could be expanded to include more details on the bioinformatics methods used. For example, the criteria for sequence quality control, taxonomy assignment, and the statistical methods used to compare the microbiota compositions between groups should be clearly stated.

13. The description of the histological scoring system is vague. It would be beneficial to provide more detailed criteria for the scoring of hepatic and colonic tissue damage, as well as examples or references to standard protocols.

14. Many grammatical errors in the text throughout the manuscript were understandable. Thus, the language of the paper should be checked carefully again.

Reviewers' comments:

Reviewer's Responses to Questions

**Comments to the Author**

1. Is the manuscript technically sound, and do the data support the conclusions?

Reviewer #1: Partly

2. Has the statistical analysis been performed appropriately and rigorously? 

Reviewer #1: Yes

3. Have the authors made all data underlying the findings in their manuscript fully available?

Reviewer #1: No

4. Is the manuscript presented in an intelligible fashion and written in standard English?

Reviewer #1: Yes

5. Review Comments to the Author

Reviewer #1: This study investigates the potential of Lactobacillus plantarum KAD supplementation to mitigate metabolic disorders, particularly in the context of high-fat diet (HFD)-induced insulin resistance (IR) and associated hepatic injuries in mice. The manuscript has several promising aspects, but it requires revisions as several points need to be carefully revised before the next resubmission as follows:

1) Clarify the hypothesis of the study at the end of the Introduction section.

2) How did you calculate the sample size for the animals included in this study?

3) What is dose of L. plantarum per kg diet?

4) The authors had to justify the selection of L. plantarum dose, supported by appropriate reference.

5) In the Results section, remove the data from the text as it is already displayed in Figures.

6) The formulation or nutrient composition of the experimental diet should be given.

7) English editing of the manuscript is highly recommended.

6. PLOS authors have the option to publish the peer review history of their article (what does this mean?). If published, this will include your full peer review and any attached files.

Reviewer #1: No

---

## [Editor Report · Decision Letter 1]

28 Oct 2024

Lactobacillus plantarum KAD protects against High-Fat Diet-Induced Hepatic Complications in Swiss Albino Mice: Role of Inflammation and Gut Integrity

PONE-D-24-27060R1

Dear Dr. Saha,

We’re pleased to inform you that your manuscript has been judged scientifically suitable for publication and will be formally accepted for publication once it meets all outstanding technical requirements.

Kind regards,

Ahmed Khalafallah Sadeq Rashwan, PhD

Academic Editor

PLOS ONE

Additional Editor Comments (optional):

Dear Dr. Tilak Saha and Co-authors,

We are pleased to inform you that your manuscript titled “Lactobacillus plantarum KAD Protects against High-Fat Diet-Induced Hepatic Complications in Swiss Albino Mice: Role of Inflammation and Gut Integrity” has been accepted for publication in PLOS ONE.

The reviewers and editors commend the thorough research and insightful findings you have presented. Your study provides important contributions to our understanding of gut microbiota’s role in mitigating hepatic complications associated with high-fat diets, particularly through the mechanisms of inflammation and gut integrity. This work holds significant implications for future research in microbiome therapy and metabolic health.

We appreciate the time and effort you invested in addressing all comments and suggestions provided by our reviewers. Your diligent revisions have further strengthened the clarity and impact of the study.

Please ensure that all final revisions, if any, are submitted in a timely manner for the next steps in the publication process. We look forward to seeing your work reach the scientific community and inspire further research in this promising area.

Thank you for choosing PLOS ONE as the platform to publish your work.
---

## [Editor Report · Acceptance letter]

1 Nov 2024

PONE-D-24-27060R1 

PLOS ONE

Dear Dr. Saha, 

I'm pleased to inform you that your manuscript has been deemed suitable for publication in PLOS ONE. Congratulations! Your manuscript is now being handed over to our production team.

Kind regards, 

on behalf of

Dr. Ahmed Khalafallah Sadeq Rashwan 

Academic Editor

PLOS ONE